# Nonoperative treatment versus volar locking plating for distal radius fracture in patients aged 65 years or older (DRIFT trial): A randomized controlled trial

Teemu P. Hevonkorpi[1,2]*, Antti P. Launonen[1], Aleksi Reito[1],
Mette Schandorff Skjærbæk[3], Yan Li[4], Toni Luokkala[5], Juha Kukkonen[6], Juha Paloneva[7],
Helle Kvistgaard Østergaard[3], Li Felländer-Tsai[4], Minna K. Laitinen[8],
Bakir O. Sumrein[1,2], Inger Mechlenburg[9], Ville M. Mattila[1,2], as the NITEP Group
(Nordic Innovative Trial to Evaluate osteoPorotic fractures)

1 Department of Orthopaedics, Tampere University Hospital, Tampere, Finland, 2 Faculty of Medicine and Health Technology, Tampere University, Tampere, Finland, 3 Department of Orthopedics, Viborg Regional Hospital, Viborg, Denmark, 4 Department of Orthopedics, Karolinska University Hospital, Huddinge, Sweden, 5 Department of Hand Surgery, Central Finland Hospital Nova, Jyväskylä, Finland, 6 Department of Surgery, Satakunta Central Hospital, Pori, Finland, 7 Institute of Clinical Medicine, School of Medicine, Faculty of Health Sciences, University of Eastern Finland, Kuopio, Finland, 8 Department of Orthopaedics, Helsinki University Hospital, Helsinki, Finland, 9 Department of Clinical Medicine, Aarhus University, Aarhus, Denmark

* teemu.hevonkorpi@tuni.fi

## Abstract

### Background

The optimal management of distal radius fractures (DRFs) in older patients remains debatable. A knowledge gap exists concerning how to manage fractures with early malalignment during nonoperative treatment. We conducted a prospective, multicenter, and randomized controlled trial to compare nonoperative treatment to operative treatment with volar locking plating (VLP) in the management of primarily malaligned DRFs and DRFs that exhibit early malalignment during nonoperative treatment.

### Methods and findings

The DRIFT trial was conducted at five trauma centers in Finland, Sweden, and Denmark. Patients aged 65 years or older with a dorsally displaced AO-type A or C DRF were included in the trial. Patients with DRF who did not maintain alignment after closed reduction (CR) were randomized in 1:1 ratio to nonoperative treatment or VLP. Patients with acceptable alignment after CR visited the outpatient clinic 5–10 days after CR. If the alignment was lost, the patients were randomized in 1:1 ratio to either continue nonoperative treatment or to VLP. The primary outcome measure was Patient Rated Wrist Evaluation (PRWE) at 12 months. The primary analysis

**Data availability statement:** Study data cannot be shared publicly because of confidentiality requirements under Finnish, Swedish, and Danish Data Security legislation. The de-identified patient data are available from the NITEP-group and from Tampere University Hospital (contact via http://nitep.eu/en/contact/ or www.pirha.fi, kirjaamo@pirha.fi) for researchers who meet the criteria to access confidential data.

**Funding:** This trial was supported by Academy of Finland (Terveyden tutkimuksen toimikunta), www.aka.fi; ref 308203, to VM; Sigrid Juselius Foundation Finland, www.sigridjuselius.fi, to VM; Finnish Insurance Companies grant, www.suomenvakuutusyhdistys.fi, to VM, Finnish State Research Funding; ref B1810 and B1910, to TL, Marshall Foundation Denmark, www.marshallsfond.dk, to IM and ALF-grant Sweden, www.intramed.lu.se/en/research/alf-funding, to YL. The funders had no role in the study design, data collection and analysis, decision to publish, or manuscript preparation.

**Competing interests:** The authors have declared that no competing interests exist.

**Abbreviations:** 15-Dfifteen dimensional quality of life questionnaire; AE, adverse events; CI, confidence interval; CONSORT, Consolidated Standards of Reporting Trials; CR, closed reduction; DRF, distal radius fracture; DRIFT, distal radius fracture trial, ER, emergency room; ITT, intention to treat; MCID, minimal clinically important difference; NITEP, Nordic Innovative Trial to Evaluate osteoPorotic fractures; PCS, pain catastrophizing scale; PRWE, patient rated wrist evaluation; QuickDASH, Quick Disabilities of Arm, Shoulder and Hand; RCTs, randomized controlled trials; SAE, serious adverse event; SD, standard deviation; VAS, visual analogue scale; VLP, volar locking plating.

method for PRWE was a linear mixed model. In the linear mixed model, patient was a random factor and age and intra/extra articularity of the fracture were fixed. Participants and orthopedic investigators were not blinded. The statisticians and investigators responsible for the analysis remained blinded to the treatment groups during data analysis and the drawing of conclusions. Between March 15, 2018 and June 6, 2023, 291 patients aged 65 years or older (mean age 73 years (standard deviation (SD) 5.8), 258 women, 33 men) who had sustained a DRF were included in the trial. The 12-month follow-up was completed on June 11, 2024. One hundred twenty-four DRFs did not maintain acceptable fracture alignment after CR; 66 were randomized to nonoperative treatment and 58 to VLP. These patients are referred to as primarily malaligned DRFs. Eighty-six patients lost fracture alignment during the first 5–10 days of follow-up; 44 patients were randomized to continue nonoperative treatment, and 42 patients to VLP. These patients are referred to as early malaligned DRFs. In primarily malaligned DRFs, the estimated mean effect for PRWE at 12 months was −9.6 points (95% confidence interval (CI) [−17.4, −1.7]; $p = 0.0178$) in favor of VLP, which is smaller than the predefined minimal clinically important difference (MCID) of the PRWE (11 points). In early malaligned DRFs, the mean effect for PRWE at 12 months was −6.2 points (95% CI [−15.4, 3.0]; $p = 0.1816$). At 12-month follow-up, we found 25 treatment-related adverse events (AE) (10/66, 15% in primarily malaligned DRFs nonoperative; 5/58, 8.6% in primarily malaligned DRFs operative; 2/44, 4.5% in early malaligned DRFs nonoperative; 3/42, 7.1% in early malaligned DRFs operative; 5/63, 7.9% in well-aligned DRFs) and 11 other AE. The trial recruitment period was longer than expected due to the restrictions caused by the global COVID-19 pandemic. Due to the decreased inclusion rate, we had to cease the recruitment of patients with early malaligned DRF before reaching the predefined 57 patients per Arm, which was the main limitation of the trial.

## Conclusions

Operative treatment of primarily malaligned DRF with VLP may slightly improve wrist function at 12 months. The estimated mean difference between the groups was, however, smaller than the predefined MCID of the PRWE (11 points). In DRFs with early loss of alignment, operative treatment does not appear to provide benefit. Our results suggest that the choice of treatment modality should be made following primary fracture reduction because subsequent monitoring of fracture alignment does not offer any additional benefit in terms of expected wrist function. This questions the need for early radiographic follow-up during the nonoperative treatment.

## Trial registration

The trial was registered at ClinicalTrials.gov (Identifier: NCT02879656, registration date 08/17/2016).

## Author summary

### Why was this study done?

• Distal radius fracture (DRF) is the most common upper limb fracture, and it is especially common in older patients.

• Previous studies comparing operative and nonoperative treatment of DRF have yielded controversial results.

• There is a knowledge gap about how we should manage DRFs that lose fracture alignment during the cast treatment.

### What did the researchers do and find?

• We randomly allocated 124 patients with DRF who had lost alignment immediately after closed reduction (CR) of the fracture to either cast treatment or to surgery with volar plate fixation (VLP). Eighty-six patients with good fracture alignment after CR lost alignment during the early follow-up and were also randomly allocated to continue cast treatment or to fixation with a volar plate.

• We evaluated patient-rated wrist evaluation score (PRWE), other functional outcomes, pain catastrophizing, radiographs, complications, and quality of life at 3 months and 12 months after the DRF.

• If DRF alignment was lost immediately after CR, surgery provided benefit in functional outcomes at 12 months. However, the mean effect was below the previously established minimal clinically important difference. If fracture alignment was lost during the early follow-up, surgery did not provide clinically important benefit.

### What do these findings mean?

• Some older patients with fracture malalignment right after CR of DRF benefit from surgery with VLP, whereas, surgery in case of fracture malalignment during the cast treatment does not provide clinically significant benefit.

• The decision whether to operate a DRF or not, should be made after CR of the fracture, and after this there is no additional benefit in controlling the fracture alignment with radiographs. This diminishes the need for follow-up visits and helps to allocate resources better.

• The main limitation of the current trial was the termination of patient recruitment in early malaligned fractures before the intended group size was reached.

## Introduction

Distal radius fracture (DRF) is the most common fracture in older patients [1–5]. DRF not only causes disability due to pain and reduced functional capacity but also has a substantial economic impact on healthcare systems due to the costs of primary treatment, follow-up, and rehabilitation [6–11].

Volar locking plate (VLP) gained immense popularity after its introduction in the early 2000s [12,13]. Since then, the main treatment methods for DRFs have been closed reduction and immobilization with cast (CR) and operative treatment with VLP [13–15].

Previous randomized controlled trials (RCTs) comparing operative and nonoperative treatment of DRF have yielded controversial results. While most studies have demonstrated the benefits of operative treatment during the early stages of recovery, the benefits have been found to diminish by 12-month follow-up [16–22]. Some RCTs have, however, also revealed a significant benefit of VLP fixation at 12-month follow-up [23,24]. Previous studies have focused on primarily malaligned fractures, and existing RCT-level evidence does not address the management of those DRFs that lose alignment during nonoperative treatment. Furthermore, most studies that have evaluated the effect of the operative treatment of DRFs that lose alignment during nonoperative treatment have been retrospective. In their systematic review and meta-analysis, Khan and colleagues concluded that delay of surgery of more than 2 weeks after DRF may be associated with inferior outcomes [25]. To our knowledge, no previous RCTs have had separate randomization and separate study arms to evaluate treatment effect between operative and nonoperative treatment in patients with early loss of alignment of DRF.

The primary aim of this prospective, multicenter, multinational RCT was to compare nonoperative treatment with VLP in the treatment of both primarily malaligned DRFs and DRFs where alignment is lost during the first 5–10 days of follow-up in patients aged 65 years and older in terms of function, pain, disability, quality of life, and complications. The primary outcome measure was PRWE at 12 months.

## Methods

### Ethics statement

The DRIFT (Distal Radius Fracture Trial) is a prospective, multicenter, multinational, randomized controlled trial conducted by the NITEP (Nordic Innovative Trial to Evaluate Osteoporotic Fractures) group. The trial protocol was approved by the Regional Ethics Committee of Tampere University Hospital, Finland (ETL-code R16105), the Scientific Ethics Committee of the Central Denmark Region, Denmark (case number 1-10-72-250-17), and the Swedish Ethical Review Authority, Sweden (case number 2019-03548). Research permits were obtained from the local ethics committees and hospital districts prior to the commencement of the trial. The trial was conducted in accordance with the Declaration of Helsinki. All patients gave informed consent prior to participation. Independent steering and monitoring committees observed the trial. The trial was registered at ClinicalTrials.gov (Identifier: NCT02879656, registration date 08/17/2016), and the trial protocol was published simultaneously with the onset of the trial [26]. The study is reported according to the recommendations of the CONSORT 2025 (S1 Checklist).

### Study design and participants

Patients with a DRF that did not maintain acceptable alignment after CR were randomized in a 1:1 ratio to either nonoperative treatment (Arm 1) or operative treatment with VLP (Arm 2). These patients comprised the primarily malaligned DRFs group.

Patients with acceptable alignment after CR were followed up for potential early malalignment. If the alignment was lost at 5–10 days of follow-up, the patients were randomized in a 1:1 ratio to nonoperative treatment (Arm 3N) or operative treatment with VLP (Arm 3O). These patients comprised the early malaligned DRFs group.

Patients with acceptable alignment after CR that maintained the alignment at 5–10 days of follow-up, continued the standard follow-up visits for nonoperative treatment (Arm 4). These patients comprised the well-aligned DRFs group.

Patients who declined randomization were asked to participate in an external follow-up group. They were treated according to local clinical practice, but had follow-up visits and questionnaires as the allocated patients.

The DRIFT trial was conducted in five trauma centers in Finland, Sweden, and Denmark (S7 Text). All centers are primary regional referral centers for orthopedic trauma patients in their area.

The inclusion criteria of the trial were low-energy intra- or extra-articular dorsally displaced DRF (AO classification type A or C [27]) within 3 cm of the radiocarpal joint diagnosed with lateral and posterior-anterior radiographs in patient aged 65

years or older. The radiologic inclusion criteria were more than 10° dorsal tilt and/or more than 2 mm step-off and/or more than 3 mm shortening in radiographs.

The exclusion criteria of the trial were refusal to participate in the study, open fracture more than Gustilo Grade 1 [28], Chauffeur's, Barton's, or Smith's fracture, inability to understand written and spoken guidance in local languages and pathological fracture or previous fracture in the same wrist or forearm.

### Randomization and masking

Patients were randomized in a block allocation fashion. Blocks of 10 were stratified by age (65–74 years and 75 years or older), sex, and intra-articular versus extra-articular fracture. The treatment allocations from the matrix were acquired from an online randomization system (http://randomize.net), where the researcher logged on after written consent was obtained and received the allocated intervention. The researchers did not have access to the allocation sequence. Physicians responsible for the interventions did not participate in collecting the primary outcomes during the follow-up.

Participants and orthopedic investigators were not blinded because the interventions were so different. The statisticians and investigators responsible for the analysis remained blinded to the treatment groups during data analysis and the drawing of conclusions.

### Procedures

All patients visiting the ER with a dorsally displaced AO-type A or C DRF underwent CR under local infiltration anesthesia into the fracture hematoma. After CR, the wrist was immobilized in a dorsal functional position cast, and fracture alignment was verified with radiographs.

Patients with primarily malaligned DRF randomized to nonoperative treatment (Arm 1), patients with early malaligned DRF randomized to continue nonoperative treatment (Arm 3N), and patients with well-aligned DRF (Arm 4) received standard nonoperative treatment with a dorsal functional position cast for 5 weeks.

Patients with primarily malaligned DRF randomized to operative treatment (Arm 2), and patients with early malaligned DRF randomized to operative treatment (Arm 3O) received surgery with VLP. Modified Henry's approach was used. As the trial was pragmatic, the same volar locking plates that the trial centers used in daily practice were used (S11 Details of volar locking plates). The wound was closed with absorbable sutures. Dorsal functional position cast was used for 2 weeks postoperatively.

After cast removal, all patients underwent an exercise program guided by a physiotherapist or occupational therapist, and the patients were given a printed exercise notebook. Detailed exercise guidelines for both groups can be found in the supplementary material (S1 Text).

### Outcomes

The primary outcome measure was the PRWE score at 12-month follow-up. The PRWE has 15 questions that assess the subjective function and pain of the wrist and hand, rated on an 11-point scale from 0 to 10, giving a total range of 0–100. The key secondary outcomes measured were QuickDASH (disabilities of the arm, shoulder, and hand), visual analogue scale (VAS) pain, Pain Catastrophizing Scale (PCS), quality of life (15-D), grip strength, frailty (Clinical Frailty Score), AE, and the number of subsequent surgeries. The Axivity accelerometer data, and the data of Edinburgh Wrist Calculator are not reported in this publication and will be reported afterwards. The 24-month results will be reported later. Detailed information of the outcome variables can be found in the supplementary material (S10 Text).

### Statistical analysis

Assuming the effect size of an 11-point difference (previously reported minimal clinically important difference (MCID) of the PRWE score) [29] in the PRWE score and a SD of 14 points, based on power calculations (CI 95%, type I error rate 5%

two-sided, power 0.95), the required sample size per Arm was 40 patients. Assuming a 30% drop-out rate based on possible operative intervention during nonoperative treatment, the group size needed was 57 per Arm. When the five Arms were considered, a total of 285 patients were needed. The randomization system included both primarily malaligned and early malaligned DRFs. Therefore, in comparison to power calculations, nine extra patients were recruited for Arm 1 and one extra patient was recruited for Arm 2 before recruitment ended in all participating centers. We had to cease the recruitment of patients with early malaligned DRF before reaching 57 patients per Arm due to a decreased inclusion rate and a change in treatment policies towards more nonoperative treatment. More detailed information can be found in the supplementary material (S4 Text).

The primary analysis used the Intention to Treat (ITT) principle. The primary analysis method for PRWE was a linear mixed model. Deviating from the published study protocol, a linear mixed model was chosen as the primary analysis method as it is currently the most common method used in longitudinal trials with a repeated measurement setup. The primary treatment effect at 12 months was estimated as the estimated marginal means between the study groups. Similar estimates were also obtained for 3 months. Moreover, a similar analysis was performed for the key secondary outcomes: QuickDASH, VAS pain, grip strength, and 15-D. Results are presented with 95% CIs. No adjustment for multiple testing from the two main comparisons was made. We estimated group absolute risk difference using logistic regression for categorical variables. Detailed statistical analysis is described in the statistical analysis plan (S2 Text).

## Results

Between March 15, 2018 and June 6, 2023, 291 patients aged 65 years or older who had sustained a DRF were included in the trial. The 12-month follow-up was completed on June 11, 2024.

Primarily malaligned DRFs: 124 patients did not fulfill the criteria for acceptable fracture alignment after CR; 66 patients were randomized to nonoperative treatment (Arm 1) and 58 patients were randomized to operative treatment with VLP (Arm 2).

Early malaligned DRFs: 149 patients fulfilled the criteria for acceptable fracture alignment after CR. Of these, 86 patients lost alignment during the first 5–10 days of follow-up; 44 patients were randomized to continue nonoperative treatment (Arm 3N) and 42 patients were randomized to operative treatment with VLP (Arm 3O).

Well-aligned DRFs: 63 patients had acceptable fracture alignment at the 5–10-day follow-up visit and continued nonoperative treatment in Arm 4.

In addition, 18 patients eligible for the trial declined the randomization and were assigned to an external follow-up group.

A CONSORT flow diagram is presented in Fig 1. The baseline characteristics are presented in Table 1.

### Primary outcome

**Primarily malaligned DRF.** The mean PRWE at 12-month follow-up was 9.7 points (SD 12.2) for patients operated with VLP and 18.7 (SD 24.6) for nonoperatively treated patients. The mean effect for 12 months PRWE was −9.6 points (95% CI [−17.4, −1.7]; $p = 0.0178$) favoring VLP. Results for primary outcome are presented in Fig 2 and Table 2. Subgroup analysis by age can be found in the supplementary material (S5 Text).

**Early malaligned DRF.** In patients with early loss of alignment operated with VLP, the mean PRWE at 12 months was 9.3 points (SD 15.7). In patients with early loss of alignment treated nonoperatively, the mean PRWE at 12 months was 16.6 points (SD 18.3). The mean effect at 12 months for PRWE was −6.2 points (95% CI [−15.4, 3.0]; $p = 0.1816$). Results for primary outcome are presented in Fig 2 and Table 2.

### Secondary outcomes

**Primarily malaligned DRF.** The mean QuickDASH at 12 months was 11.3 points (SD 12.5) for patients operated with VLP and 19.4 points (SD 22.2) for nonoperatively treated patients. The mean effect at 12 months for QuickDASH was

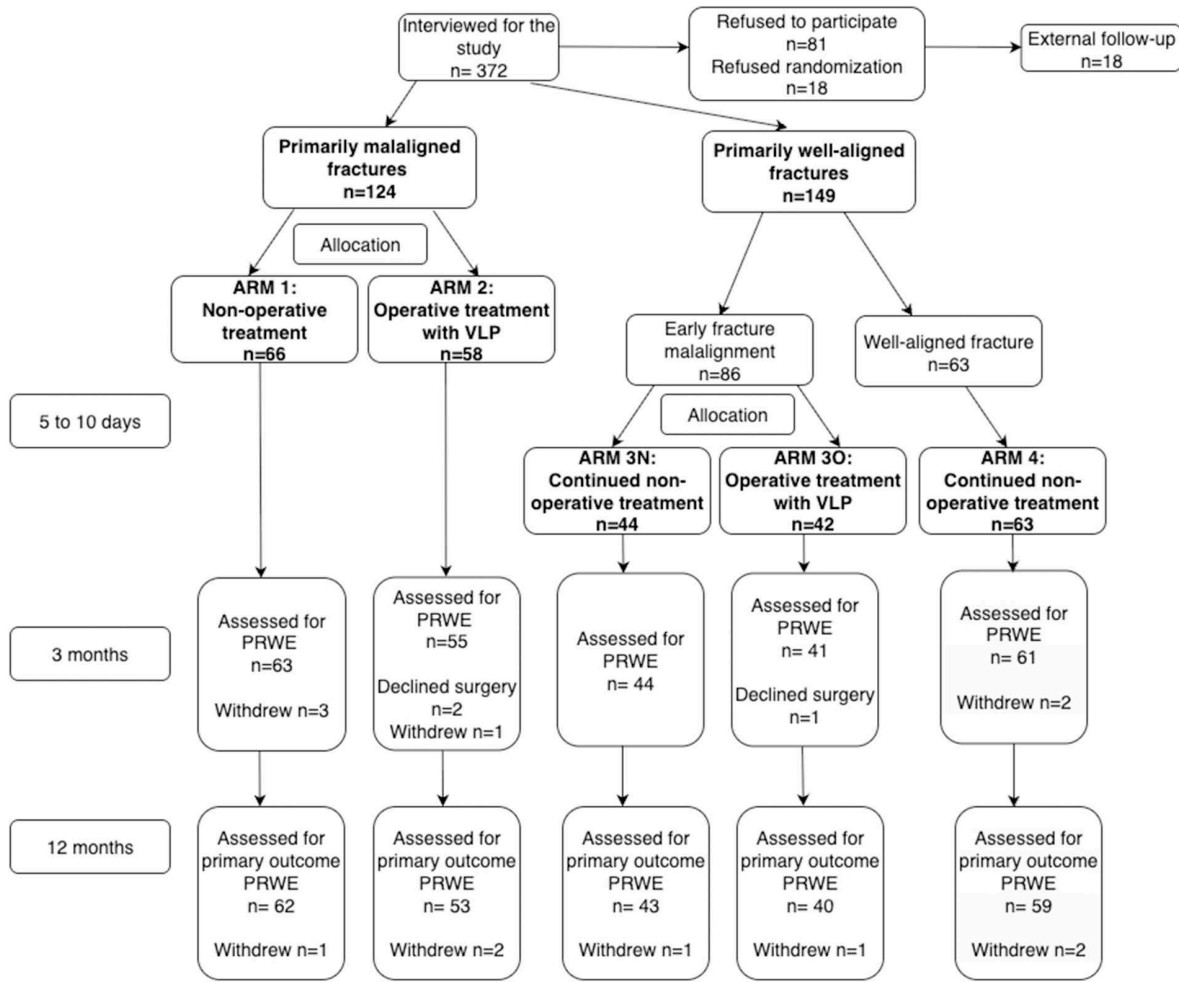

**Fig 1. Participant flow diagram of the DRIFT trial.** All patients were analyzed according to ITT analysis. DRIFT, distal radius fracture trial; ITT, intention to treat; PRWE, Patient-Rated Wrist Evaluation; VLP, volar locking plating.

−9.3 points (95% CI [−16.3, −2.3]) favoring VLP. At 12 months, the mean effect for grip strength was 5.3 kg (95% CI [2.9, 7.7]) and for 15-D 0.017 (95% CI [−0.001, 0.04]), both exceeding the level of MCID. Detailed results for other secondary outcomes are presented in Table 2.

**Early malaligned DRF.** In patients with early loss of alignment operated with VLP, the mean QuickDASH at 12 months was 11.1 points (SD 14.3). In patients with early loss of alignment treated nonoperatively, the mean QuickDASH at 12 months was 15.4 points (SD 15.2). The mean effect at 12 months for QuickDASH was −3.0 points (95% CI [−11.4, 5.3]). At 12 months, the mean effect for grip strength was 5.3 kg (95% CI [3.2, 7.4]) reaching the level of MCID.

## Adverse events

At 12-month follow-up, we found 25 treatment-related AE (10/66, 15% in primarily malaligned DRFs nonoperative; 5/58, 8.6% in primarily malaligned DRFs operative; 2/44, 4.5% in early malaligned DRFs nonoperative; 3/42, 7.1% in early malaligned DRFs operative; 5/63, 7.9% in well-aligned DRFs) and 11 other AE. All AE are presented in Table 3 and in detail in supplementary material (S3 Text).

PLOS Medicine

**Table 1. Baseline characteristics of 291 patients by randomization group.**

| Treatment groups | Arm 1 (N=66) (primarily malaligned DRF, nonoperative) | Arm 2 (N=58) (primarily malaligned DRF, operative) | Arm 3N (N=44) (early malaligned DRF, nonoperative) | Arm 3O (N=42) (early malaligned DRF, operative) | Arm 4 (N=63) (well-aligned DRF, nonoperative) | External Group (N=18) |
|---|---|---|---|---|---|---|
| Mean age (SD, Range) | 74 years (SD 5.7, range 65–88) | 72 years (SD 5.7, range 65–88) | 72 years (SD 5.2, range 65–86) | 73 years (SD 6.0, range 65–89) | 73 years (SD 5.9, range 65–86) | 75 years (SD 6.4, range 67–89) |
| Female sex, N (%) | 55/66 (83%) | 54/58 (93%) | 38/44 (86%) | 40/42 (95%) | 54/63 (86%) | 17/18 (94%) |
| Hand dominance right, N (%) | 62/66 (97%) | 55/58 (95%) | 41/44 (93%) | 39/42 (93%) | 60/63 (95%) | 14/18 (78%) |
| Fracture on dominant side, N (%) | 27/66 (41%) | 29/58 (50%) | 17/44 (39%) | 19/42 (45%) | 32/63 (51%) | 4/18 (22%) |
| AO-type A fracture, N (%) | 24/66 (36%) | 18/58 (31%) | 17/44 (39%) | 20/42 (48%) | 36/63 (57%) | 4/18 (22%) |
| AO-type C fracture, N (%) | 42/66 (64%) | 40/58 (69%) | 27/44 (61%) | 22/42 (52%) | 27/63 (43%) | 14/18 (78%) |
| Time from injury to operation, mean (range), days* | NA | 7 (1-18) | NA | 12 (7-24) | NA | 7 (7) |
| Smoking, N (%) | 2/66 (3%) | 10/58 (17%) | 4/44 (9%) | 4/42 (10%) | 3/63 (5%) | 3/18 (17%) |
| Diabetes, N (%) | 4/66 (6%) | 6/58 (10%) | 3/44 (7%) | 7/42 (17%) | 4/63 (6%) | 3/18 (17%) |
| Frailty**, mean (SD) | 1.9 (SD 1.1) | 1.6 (SD 0.7) | 1.6 (SD 0.9) | 1.2 (SD 0.6) | 1.5 (SD 0.7) | NA |
| Osteoporosis***, N (%) | 3/66 (5%) | 6/58 (10%) | 3/44 (7%) | 4/42 (10%) | 4/63 (6%) | 4/18 (22%) |
| Peroral corticosteroid medication, N (%) | 5/66 (8%) | 3/58 (4%) | 3/44 (7%) | 1/42 (2%) | 3/63 (5%) | 1/18 (6%) |
| Biological medication, N (%) | 1/66 (2%) | 0/58 (0%) | 1/44 (2%) | 0/42 (0%) | 6/63 (10%) | 1/18 (6%) |

Data represent number (percentage) unless otherwise indicated. *Time between injury and operative treatment reported as mean number of days (range).

**Clinical Frailty Score, range from 1 to 9, 1 indicating physically active, fit and energetic senior citizen, 9 indicating very frail person with life expectancy <6 months.

***Osteoporosis diagnosed before the distal radius fracture.

*Abbreviations:* DRF, distal radius fracture; SD, standard deviation.

## Discussion

In this prospective randomized clinical trial, we found that operative treatment of primarily malaligned DRFs may improve wrist function measured with PRWE at 12-month follow-up. The mean difference between the groups was, however, smaller than the predefined MCID of the PRWE (11 points). In patients with primarily acceptable fracture reduction and early malalignment at 5–10-day follow-up, operative treatment does not provide improvement in wrist function measured with PRWE at 12 months.

The mean effect for PRWE at 12 months was −9.6 points (95% CI [−17.4, −1.7]; p = 0.0178) in favor of operative treatment in patients with primarily malaligned DRF. This finding supports the findings of RCTs conducted by Saving and colleagues, Selles and colleagues, and Martinez-Mendez and colleagues [19,23,24]. Previous meta-analyses have reported a minor benefit for operative treatment, but the effect sizes have generally been reported to be smaller than the MCID of the other commonly used upper limb patient-reported outcome measure, the DASH [30–33]. In their recent meta-analysis of 12 RCTs comparing volar plating and the nonoperative treatment of DRFs, Linnanmäki and colleagues found that even

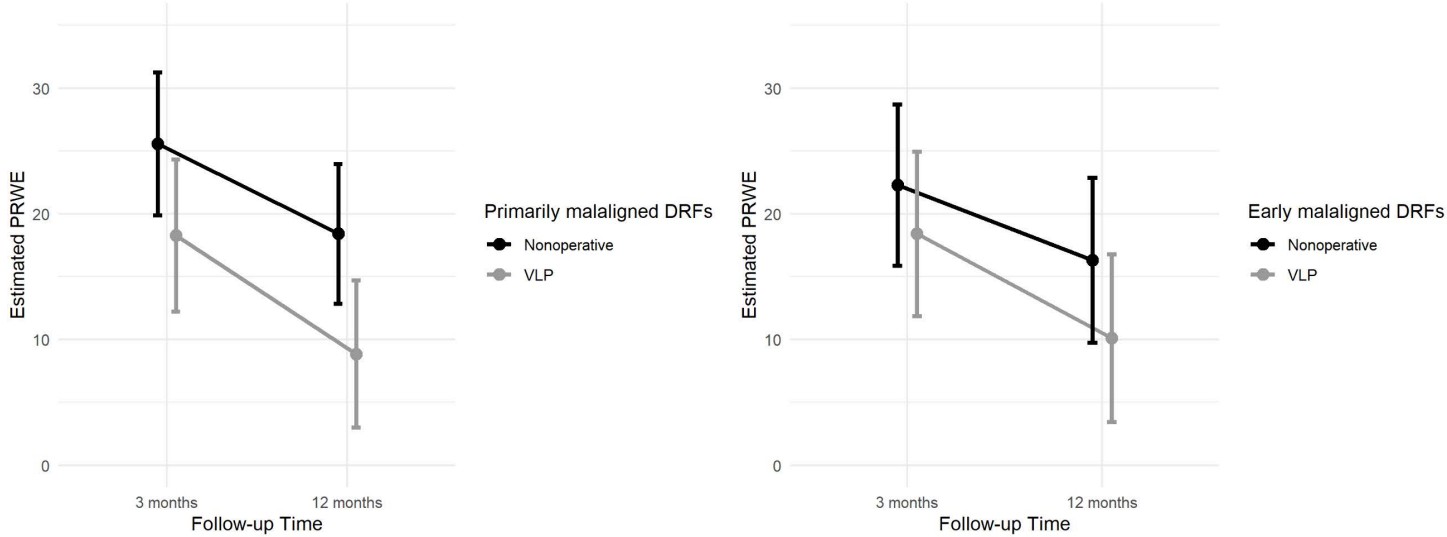

**Fig 2. Between-group differences of mean PRWE with 95% confidence intervals in primarily malaligned DRFs and early malaligned DRFs at 3-month and 12-month follow-up.** PRWE is scaled from 0 to 100 with the minimum (=best) score of 0. DRF, distal radius fracture; PRWE, Patient-Rated Wrist Evaluation; VLP, volar locking plating.

though the mean effects tended to be smaller than the MCID of the DASH, the nonoperatively treated groups had wider variances; meaning that they had more patients with poor outcomes. The risk for poor outcome was 10–15% lower with VLP regardless of which cutoff in DASH was chosen to present poor outcome [33]. Supporting these findings, the SD for 12-month PRWE in the present study was 24.6 in patients with primarily malaligned DRF treated nonoperatively and 12.2 in patients operated with VLP.

A key gap in the previous literature has been the management of those DRFs that lose alignment during nonoperative treatment. Specifically, it has been unclear whether performing an operation with VLP in these cases provides benefit in terms of wrist function. In our study, 44 patients continued nonoperative treatment after fracture alignment was lost during the first 5–10 days of follow-up, and 42 patients were operatively treated with VLP. The mean effect for 12-month PRWE was −6.2 points (95% CI [−15.4, 3.0]; $p = 0.1816$), whereas in QuickDASH the mean effect was −3.0 points (95% CI [−11.4, 5.3]). It seems that if the fracture alignment is lost during nonoperative treatment, surgical intervention with VLP provides no improvement in wrist function measured with PRWE at 12-month follow-up, and thus questions the benefit of follow-up appointments after the initial treatment modality has been selected. In contrast to the previously published RCTs on DRFs, the current trial had separate randomization and separate study arms to evaluate treatment effect between operative and nonoperative treatment in patients with early loss of alignment of DRF.

Previously published RCTs have focused on primarily malaligned DRFs and demonstrated that operative treatment is beneficial in the early stages of recovery. However, the differences between operative and nonoperative methods diminish over time [16–22]. Our results partially contradict these findings, since primarily operatively treated patients in our trial also had better function in PRWE and QuickDASH scores at 12-month follow-up. Even though the mean effect was smaller than the predefined MCID, the benefit of operative treatment was consistent throughout the follow-up and the 95% CIs also included clinically significant differences. Mulders and colleagues have previously reported that the normative value of PRWE in patients aged 65 years and older is 9.2 (SD 16.0) [34]. Overall, the participants in the present trial achieved good wrist function at 12-month follow-up, and the results of 12-month PRWE were near the normative value in all groups.

**Table 2. Primary and secondary outcomes of the randomized clinical trial groups at 3 months and 12 months.**

| Treatment groups | Arm 1 (primarily malaligned DRF, nonoperative) mean (SD) | Arm 2 (primarily malaligned DRF, operative) mean (SD) | Arm 2 vs. Arm 1 mean difference, 95% CI | Arm 3N (early malaligned DRF, nonoperative) mean (SD) | Arm 3O (early malaligned DRF, operative) mean (SD) | Arm3O vs. Arm3N mean difference, 95% CI | Arm 4 (well-aligned DRF, nonoperative) mean (SD) | External Group, mean (SD) |
|---|---|---|---|---|---|---|---|---|
| **3 months** | | | | | | | | |
| PRWE | 25.8 (SD 21.8) | 19.7 (SD 20.1) | −7.3 (95% CI [−15.4, 0.8]) | 23.8 (SD 18.5) | 18.4 (SD 19.6) | −3.9 (95% CI [−12.9, 5.1]) | 22.6 (SD 17.6) | 35.8 (SD 24.9) |
| QuickDASH | 25.7 (SD 17.7) | 20.5 (SD 18.7) | −5.7 (95% CI [−12.7, 1.3]) | 25.8 (SD 17.0) | 20.9 (SD 18.0) | −4.6 (95% CI [−12.8, 3.26]) | 19.3 (SD 16.4) | 37.0 (SD 16.1) |
| 15-D | 0.91 (SD 0.07) | 0.93 (SD 0.07) | 0.023 (95% CI [−0.004, 0.05]) | 0.91 (SD 0.07) | 0.91 (SD 0.10) | −0.004 (95% CI [−0.04, 0.04]) | 0.92 (SD 0.07) | 0.85 (SD 0.11) |
| Grip strength difference, kg* | −13.6 (SD 5.9) | −7.6 (SD 5.5) | 6.4 (95% CI [4.1, 8.7]) | −13.3 (SD 4.6) | −7.4 (SD 4.3) | 5.3 (95% CI [3.3, 7.4]) | −10.2 (SD 6.4) | −10.0 (SD 6.9) |
| VAS pain, cm | 1.8 (SD 1.9) | 1.5 (SD 2.0) | −0.3 (95% CI [−0.8, 0.2]) | 2.1 (SD 2.0) | 1.7 (SD 2.1) | −0.4 (95% CI [−1.1, 0.3]) | 1.7 (SD 1.6) | 3.5 (SD 2.7) |
| PCS | 17.6 (SD 8.0) | 17.7 (SD 6.4) | NA | 19.1 (SD 7.2) | 18.4 (SD 9.5) | NA | 16.7 (SD 6.2) | 21.0 (SD 13.3) |
| **12 months** | | | | | | | | |
| PRWE | 18.7 (SD 24.6) | 9.7 (SD 12.2) | −9.6 (95% CI [−17.4, −1.7]) | 16.6 (SD 18.3) | 9.3 (SD 15.7) | −6.2 (95% CI [−15.4, 3.0]) | 11.9 (SD 15.4) | 24.0 (SD 21.2) |
| QuickDASH | 19.4 (SD 22.2) | 11.3 (SD 12.5) | −9.3 (95% CI [-16.3, −2.3]) | 15.4 (SD 15.2) | 11.1 (SD 14.3) | −3.0 (95% CI [−11.4, 5.3]) | 14.3 (SD 15.0) | 21.4 (SD 19.6) |
| 15-D | 0.91 (SD 0.07) | 0.93 (SD 0.08) | 0.017 (95% CI [−0.001, 0.04]) | 0.91 (SD 0.08) | 0.92 (SD 0.09) | −0.004 (95% CI [−0.04, 0.04]) | 0.93 (SD 0.07) | 0.87 (SD 0.11) |
| Grip strength difference, kg* | −6.9 (SD 5.4) | −1.6 (SD 5.8) | 5.3 (95% CI [2.9, 7.7]) | −8.0 (SD 5.2) | −2.3 (SD 3.5) | 5.3 (95% CI [3.2, 7.4]) | −4.9 (SD 5.1) | NA |
| VAS pain, cm | 1.0 (SD 1.4) | 0.8 (SD 1.3) | −0.2 (95% CI [−0.7, 0.3]) | 1.4 (SD 1.9) | 0.7 (SD 1.3) | −0.7 (95% CI [−1.5, −0.1]) | 1.1 (SD 1.7) | 2.1 (SD 2.3) |
| PCS | 20.0 (SD 10.9) | 18.0 (SD 7.9) | NA | 17.8 (SD 7.9) | 17.9 (SD 9.6) | NA | 16.9 (SD 7.6) | 23.2 (SD 11.6) |
| Dorsal angulation angle, degrees | 17.9 (SD 14.2) | −3.8 (SD 8.3) | −21.7 (95% CI [−26.0, −17.5]) | 18.1 (SD 12.6) | 0.3 (SD 9.5) | −17.9 (95% CI [−22.8, −13.0]) | 5.9 (SD 10.1) | 17.2 (SD 14.3) |
| Inclination angle, degrees | 15.1 (SD 6.6) | 22.6 (SD 4.3) | 7.5 (95% CI [5.5, 9.5]) | 17.0 (SD 5.3) | 22.3 (SD 4.7) | 5.4 (95% CI [3.2, 7.6]) | 19.0 (SD 7.0) | 14.6 (SD 5.9) |
| Shortening of the radius, mm | 4.1 (SD 2.8) | 0.5 (SD 2.2) | −3.6 (95% CI [−4.5, −2.7]) | 3.0 (SD 2.5) | 0.5 (SD 1.4) | −2.6 (95% CI [−3.4, −1.7]) | 2.0 (SD 1.7) | 3.2 (SD 1.6) |

Data are presented as mean (SD) for study arms and mean difference (95% confidence interval) for between-group differences.

*Grip strength reported as difference between fractured and healthy wrist in kilograms.

*Abbreviations:* C, confidence interval; DRF, distal radius fracture; PCS, Pain Catastrophizing Scale; PRWE, Patient-Rated Wrist Evaluation; QuickDASH, Quick Disabilities of Arm, Shoulder and Hand; 15-D, 15 dimensional (health-related quality of life); SD, standard deviation; VAS, Visual Analogue Scale.

In this trial, the 12-month mean effect between treatment groups for grip strength compared to the uninjured upper limb was 5.3 kg (95% CI [2.9, 7.7]) better for primarily malaligned DRFs treated with VLP and 5.3 kg (95% CI [3.2, 7.4]) better for early malaligned DRFs in favor of VLP. This exceeds the previously determined level of MCID for grip strength [35]. As the normative grip strength value for older individuals has been reported to be between 28 and 45 kg in men and between 15 and 26 kg for women, the difference of 5.3 kg may have an association with the capacity of older adults to carry out daily activities [36].

**Table 3. Adverse events at 12-month follow-up by study group.**

| Treatment groups | Arm 1 (n=66) (primarily malaligned DRF, nonoperative) | Arm 2 (n=58) (primarily malaligned DRF, operative) | Arm 3N (n=44) (early malaligned DRF, nonoperative) | Arm 3O (n=42) (early malaligned DRF, operative) | Arm 4 (n=63) (well-aligned DRF, nonoperative) | External Group (n=18) |
|---|---|---|---|---|---|---|
| **Treatment-related adverse events** | | | | | | |
| Carpal tunnel syndrome | 6/66 (9%) | 2/58 (3%) | 0 | 1/42 (2%) | 0 | 0 |
| Symptomatic malunion | 3/66 (3%) | 0 | 1/44 (2%) | 0 | 2/63 (3%) | 0 |
| Nonunion | 0 | 0 | 0 | 0 | 1/63 (2%) | 0 |
| Plate/screw loosening | 0 | 3/58 (5%) | 0 | 0 | 0 | 0 |
| CRPS | 0 | 0 | 1/44 (2%) | 0 | 0 | 0 |
| Superficial infection | 0 | 0 | 0 | 2/42 (5%) | 0 | 0 |
| Deep infection | 0 | 0 | 0 | 0 | 0 | 0 |
| Tendon rupture/ dysfunction | 0 | 0 | 0 | 0 | 2/63 (3%) | 0 |
| Pressure ulcer | 1/66 (2%) | 0 | 0 | 0 | 0 | 0 |
| **Adverse events unrelated to received treatment** | | | | | | |
| Acute myocardial infarction | 1/66 (2%) | 0 | 0 | 0 | 0 | 0 |
| Cerebral infarction/TIA | 1/66 (2%) | 1/58 (2%) | 0 | 0 | 0 | 0 |
| Other hospitalized infection | 0 | 1/58 (2%) | 0 | 1/42 (2%) | 0 | 0 |
| Other fracture | 2/66 (3%) | 0 | 2/44 (4.5%) | 0 | 1/63 (2%) | 0 |
| Death | 0 | 1/58 (2%) | 0 | 0 | 0 | 0 |

Data are presented as number (percentage). *Abbreviations:* CRPS, complex regional pain syndrome; DRF, distal radius fracture; TIA, transient ischemic attack.

Certain fracture characteristics, such as volar comminution, loss of radial inclination, shortening of the radius, and increasing patient age, have been reported to be risk factors that predispose loss of alignment of the DRF [37–42]. In our trial, patients randomized to nonoperative treatment typically had quite severe malalignment on final radiographs. In previous RCTs, the degree of malalignment has varied [16,18–21,23,24,43]. However, as older patients are generally more tolerant of malunion, predicting who would benefit from better radiologic alignment remains challenging [44–47]. Surprisingly, the mean effect of 12-month PRWE in primarily malaligned DRFs in the present trial was −9.8 points (95% CI [−20.1, 0.4]) for patients aged from 65 to 74 years and −9.3 points (95% CI [−21.8, 3.1]) for patients aged 75 years or older. This contradicts the previous finding that increasing age diminishes the benefits of the operative treatment of DRF [30,33,48,49].

The overall incidence of complications was low in this trial, and there were no significant disparities in complication rates across the various treatment groups. Indeed, only 6/291 (2%) patients had a serious adverse event during the follow-up. Considering the mean age of the study population, these findings may reflect that these patients are relatively healthier than other low-energy fracture patients, such as proximal humerus fracture and hip fracture patients [50–52].

This trial has some limitations. Due to the decreased inclusion rate, we had to cease the recruitment of patients with early malaligned DRF before reaching the predefined 57 patients per Arm. The trial recruitment period was longer than expected due to the restrictions caused by the global COVID-19 pandemic, which affected the trial for many years. Furthermore, a change in local treatment practices towards more nonoperative treatment led to more patients declining to join

the study, that could have led to undergoing surgical treatment. In addition, the trial was conducted in Finland, Sweden, and Denmark, all of which have rather good healthcare resources on a global scale. This may limit the applicability of the results in less-resourced healthcare environments. A strength of this trial is the multinational, multicenter, and pragmatic nature of the trial, which gives the results good external validity. To our knowledge, our trial is also the first RCT to compare the operative and nonoperative treatment of DRF in cases of early loss of fracture alignment during nonoperative treatment. In addition, we had very few cross-overs, which improved the reliability of the ITT analysis.

According to the results of the DRIFT trial, the operative treatment of primarily malaligned DRF with VLP fixation may improve wrist function not only during early recovery but also at 12-month follow-up, although the mean effect was smaller than the predefined MCID. In cases of loss of fracture alignment during nonoperative treatment, it seems that surgery with VLP does not provide benefit.

Our results suggest that the choice of treatment modality should be made following primary fracture reduction because subsequent monitoring of fracture alignment does not offer any additional benefit in terms of expected wrist function. This questions the need for early radiographic follow-up during the nonoperative treatment.

## Supporting information

**S1 Checklist. DRIFT TRIAL CONSORT 2025 Checklist.** This checklist is licensed under the Creative Commons Attribution 4.0 International License (CC BY 4.0; https://creativecommons.org/licenses/by/4.0/).
(DOCX)

**S2 Checklist. DRIFT TRIAL CONSERVE-checklist.**
(DOCX)

**S1 Text. DRIFT TRIAL exercise guidelines notebook.**
(DOCX)

**S2 Text. DRIFT TRIAL statistical analysis plan.**
(PDF)

**S3 Text. DRIFT TRIAL adverse events.**
(DOCX)

**S4 Text. DRIFT TRIAL deviations from the study protocol.**
(DOCX)

**S5 Text. DRIFT TRIAL additional analyses.**
(DOCX)

**S6 Text. DRIFT TRIAL patient involvement.**
(DOCX)

**S7 Text. DRIFT TRIAL participating centers.**
(DOCX)

**S8 Text. DRIFT TRIAL conclusions.**
(PDF)

**S9 Text. DRIFT TRIAL research Plan.**
(DOC)

**S10 Text. DRIFT TRIAL metadata.**
(DOCX)

**S11 Text. DRIFT TRIAL details of volar locking plates.**
(DOCX)

## Acknowledgments

We want to thank all the participants in the DRIFT trial, and the organizations and personnel who participated in the treatment of the patients. We want to thank senior statistician Sakke Purolainen for his valuable help in the statistical analysis. We want to thank research nurses Meri Järvinen, Janika Pietilä, Seija Rautiainen, and Saara-Maija Hinkkanen, occupational therapist Trine Kjeldsen, and hand therapist Anneli Nilsson Geneve for their valuable work in organizing the control visits and the daily schedules of the trial.

The Nordic Innovative Trial to Evaluate osteoPorotic fractures (NITEP) group Authors: Teemu P. Hevonkorpi, MD; Antti P. Launonen, MD, PhD; Aleksi Reito, MD, PhD; Mette Schandorff Skjærbæk, MD; Yan Li, MD, PhD, prof.; Toni Luokkala, MD, PhD; Juha Kukkonen, MD, PhD; Juha Paloneva, MD, PhD, prof.; Helle Kvistgaard Østergaard, PT, MS, PhD; Li Felländer-Tsai, MD, PhD; Minna K. Laitinen, MD, PhD; Bakir O. Sumrein, MD; Inger Mechlenburg, PT, PhD, DMS; Ville M. Mattila, MD, PhD, prof.

## Author contributions

**Conceptualization:** Teemu P. Hevonkorpi, Antti P. Launonen, Aleksi Reito, Mette Schandorff Skjærbæk, Yan Li, Toni Luokkala, Juha Paloneva, Helle Kvistgaard Østergaard, Li Felländer-Tsai, Minna K. Laitinen, Bakir O. Sumrein, Inger Mechlenburg, Ville M. Mattila.

**Data curation:** Teemu P. Hevonkorpi, Antti P. Launonen, Aleksi Reito, Juha Paloneva.

**Formal analysis:** Teemu P. Hevonkorpi, Antti P. Launonen, Aleksi Reito, Juha Kukkonen, Helle Kvistgaard Østergaard, Li Felländer-Tsai, Minna K. Laitinen, Bakir O. Sumrein, Inger Mechlenburg, Ville M. Mattila.

**Funding acquisition:** Antti P. Launonen, Yan Li, Toni Luokkala, Minna K. Laitinen, Inger Mechlenburg, Ville M. Mattila.

**Investigation:** Teemu P. Hevonkorpi, Antti P. Launonen, Aleksi Reito, Mette Schandorff Skjærbæk, Yan Li, Juha Kukkonen, Juha Paloneva, Helle Kvistgaard Østergaard, Li Felländer-Tsai, Bakir O. Sumrein, Inger Mechlenburg, Ville M. Mattila.

**Methodology:** Teemu P. Hevonkorpi, Antti P. Launonen, Aleksi Reito, Yan Li, Toni Luokkala, Juha Kukkonen, Juha Paloneva, Helle Kvistgaard Østergaard, Inger Mechlenburg, Ville M. Mattila.

**Project administration:** Teemu P. Hevonkorpi, Antti P. Launonen, Aleksi Reito, Mette Schandorff Skjærbæk, Yan Li, Toni Luokkala, Juha Kukkonen, Juha Paloneva, Helle Kvistgaard Østergaard, Li Felländer-Tsai, Minna K. Laitinen, Bakir O. Sumrein, Inger Mechlenburg, Ville M. Mattila.

**Resources:** Antti P. Launonen.

**Supervision:** Antti P. Launonen, Aleksi Reito, Mette Schandorff Skjærbæk, Yan Li, Toni Luokkala, Juha Kukkonen, Juha Paloneva, Ville M. Mattila.

**Validation:** Teemu P. Hevonkorpi, Aleksi Reito, Ville M. Mattila.

**Visualization:** Teemu P. Hevonkorpi.

**Writing – original draft:** Teemu P. Hevonkorpi, Antti P. Launonen, Aleksi Reito, Toni Luokkala, Minna K. Laitinen, Ville M. Mattila.

**Writing – review & editing:** Teemu P. Hevonkorpi, Antti P. Launonen, Aleksi Reito, Mette Schandorff Skjærbæk, Yan Li, Toni Luokkala, Juha Kukkonen, Juha Paloneva, Helle Kvistgaard Østergaard, Li Felländer-Tsai, Minna K. Laitinen, Bakir O. Sumrein, Inger Mechlenburg, Ville M. Mattila.

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
