## [Editor Report · Decision Letter 0]

23 Jun 2025

Dear Dr Hevonkorpi, 

Thank you for submitting your manuscript entitled "DRIFT trial: Nonoperative treatment vs volar locking plating for distal radius fracture in patients aged 65 years or older: A randomized clinical trial" for consideration by PLOS Medicine.

Your manuscript has now been evaluated by the PLOS Medicine editorial staff and I am writing to let you know that we would like to send your submission out for external peer review.

For clinical studies, please upload a copy of your trial study protocol as a supporting information file. The study protocol should be the version submitted for approval to the institutional review board or ethics committee, should include any amendments to the study protocol, as well as the date of their approval by the institutional review or ethics committee. Please also detail any deviations from the study protocol in the Methods section of your manuscript. The editors will consider the protocol and study conduct prior to a final decision for external review. 

Please re-submit your manuscript within two working days, i.e. by Jun 25 2025.

Feel free to email me at atosun@plos.org or us at plosmedicine@plos.org if you have any queries relating to your submission.

Kind regards,

Alexandra Tosun, PhD

Senior Editor

PLOS Medicine

---

## [Decision Letter · Decision Letter 1]

15 Jul 2025

Dear Dr Hevonkorpi,

Many thanks for submitting your manuscript "DRIFT trial: Nonoperative treatment vs volar locking plating for distal radius fracture in patients aged 65 years or older: A randomized clinical trial" (PMEDICINE-D-25-02233R1) to PLOS Medicine. The paper has been reviewed by subject experts and a statistician; their comments are included below and can also be accessed here: [LINK]

As you will see, the reviewers support a revision and provide several points of clarification and suggestions for improving the manuscript. After discussing the paper with the editorial team, I'm pleased to invite you to revise the paper in response to the reviewers' comments. We plan to send the revised paper to some or all of the original reviewers, and we cannot provide any guarantees at this stage regarding publication.

We ask that you submit your revision by Aug 05 2025. However, if this deadline is not feasible, please contact me by email, and we can discuss a suitable alternative.

Don't hesitate to contact me directly with any questions (atosun@plos.org). 

Best regards, 

Alexandra 

Alexandra Tosun, PhD 

Senior Editor

PLOS Medicine

atosun@plos.org

Comments from the editorial team:

We believe that the manuscript is potentially too complex for a nonspecialist audience. When revising it, please keep in mind that it should be accessible to a broad medical audience.

Comments from the reviewers: 

Reviewer #1: Clarity and Language

Simplify complex or repetitive sentences (e.g., in abstract and discussion).

Improve flow and readability, especially in the introduction.

Limitations and Statistical Interpretation

Explicitly discuss the risk of type II error in the early malalignment subgroup.

Consider including a post hoc power analysis or Bayesian interpretation for underpowered comparisons.

Generalizability

Acknowledge that the Nordic setting may limit applicability in less-resourced healthcare environments.

Adverse Events Reporting

Clarify which adverse events are treatment-related vs. unrelated.

Consider using a standard grading system (e.g., Clavien-Dindo) for complications.

Figures and Tables

Improve font size and clarity in the flowchart (Fig 1).

Use colorblind-friendly design and clearer axis labels in Fig 2 (e.g., note that lower PRWE = better function).

Ensure consistent naming of trial arms across text and visuals.

Reproducibility

Provide more surgical detail (plate type, reduction criteria) in the supplementary materials.

Interpretation and Conclusions

Emphasize the marginal clinical significance of VLP benefits in primarily malaligned fractures.

Highlight potential clinical implications: reduced need for early radiographic follow-up in some cases.

Abstract and Title

Clarify in the abstract that observed benefits were below the MCID.

Consider revising the title to include "randomized controlled trial" for clarity.

Consistency

Standardize age group descriptions (e.g., use consistent phrasing for ">75" vs "older than 75").

Reviewer #2: 

This manuscript reports the findings of the DRIFT trial. It is a well conducted trial and the reporting is also good. There are minor points made below but there are no major flaws in the study.

1. Introduction. The comments "existing RCT level evidence does not address the management of those DRFs that lose alignment during nonoperative treatment" and "To our knowledge, no previous RCTs have compared operative and nonoperative treatment in cases of early loss of alignment" are not correct. For example, the trial by Arora et al only included people with "inadequate initial reduction" and "loss of reduction at one week" i.e., patients who "lose alignment during non-operative treatment", which directly contradicts the statements in the Introduction. In fact, the inclusion criteria in the Arora study look the same as the current study.

2. Methods. "Gustilo 1 gradus" should be "Gustilo Grade 1" and should be referenced.

3. Methods, outcomes. Was the outcome "the number of concomitant surgeries" measuring true "concomitant" (meaning "at the same time") surgeries, or subsequent surgeries?

4. Methods. Stats. The statement "We estimated group absolute risk difference using logistic regression" seems wrong. Logistic regression provides odds ratios, not absolute risk differences.

5. Outcome scores. These needs better explanation. I have never heard of the "15-D" and there is no reference provided for this or any other outcomes, and no information on the range and whether higher scores are better or worse.

6. Discussion. I find sentences like this to be unnecessary and misleading: "Still, according to 95% confidence intervals, we cannot definitively rule out the benefit of operative treatment". It is also true that you cannot rule out the benefit of non-operative treatment (for the late malalignment group referenced). I would remove this sentence. The readers can judge the 95% CI for themselves.

7. Discussion. Similarly, I find statements like this to be misleading: "the nonoperatively treated groups had wider variances; meaning that they had more patients with poor outcomes". If they had wider variances, did they not also have more patients with very good outcomes? I think this line of argument is measuring things that were not intended to be measured, and only measuring one side of the argument is biased. I would remove this section.

8. Discussion. The authors again argue that the "key gap" in the literature is patients with loss of alignment after initial reduction. This means they have not read the inclusion criteria in the supplement of the Arora et al study that they reference. Same for their statement further down "To our knowledge, this is the first trial…" and in line 414 "Out study is the first…". This is always a dangerous thing to claim.

9. Discussion. There is too much repetition of results in the discussion. It is not necessary to repeat the kg strength measures, the 95% CIs, table references, etc in the discussion as the results have already been provided in the Results section. Talk ABOUT the results, don't repeat them unless absolutely necessary (e.g. comparing the difference found with the MCID)

10. It is not allowed to present NEW results in the discussion (like the difference in effect between age groups). This looks like it came from Supplement 6, which also includes the per protocol analysis. These things should be provided in the Results, even if it is just a reference to the supplement.

11. Discussion, line 390. It is not possible for fractures to "dislocate". Perhaps the authors meant "displace" or "lose alignment"?

12. Discussion, line 411. How did the change in local treatment "towards more non-operative treatment" lead to "more patients … undergoing surgical treatment". This seems contradictory.

Reviewer #3: Statistical review

This paper reports a trial that compared volar locking plating vs non-operative treatment at two points of a patient's treatment for distal radius fracture. I thought that the statistical methods and results were generally reported well and only had some minor comments.

1. Abstract and results: for the primary outcome, I would recommend a p-value is given in addition to confidence interval. This would seem appropriate given that the trial was powered to test a null hypothesis of no difference (and appears consistent with the SAP).

2. Abstract: although the point estimate was lower than the MCID, the uncertainty intervals do seem consistent with the MCID being plausible. I would recommend adding 'estimated' before 'mean difference' if the authors wish to retain this point.

3. Outcome variables, page 9: I note in the SAP and the CT.gov page, the plan was to include 2-year data in the analysis. Was this abandoned due to the trial finishing early, or will the 2 year results be reported elsewhere?

4. Sample size, page 9: I assume that from the '95% CI' mentioned, that the type I error rate was 5%, two-sided - I would recommend that this is added. It may also be useful to clarify that no adjustment for multiple testing from the two main comparisons was made.

5. Statistical analysis, page 9: for the linear mixed effects model, it would be useful to briefly give what fixed and random effects were used. For instance, was baseline PRWE adjusted for?

6. Statistical analysis, page 9: I would briefly mention how missing outcome data was handled (presumably it was included in the linear mixed effects model, which would be valid under a MAR assumption). The proportion of participants who were lost to follow-up was much lower than assumed, so this issue isn't too important!

7. Secondary outcomes: although all the results are presented in the table, it may be useful to have a brief narrative of which secondary results were significant and which weren't.

James Wason

---

* Please upload any figures associated with your paper as individual TIF or EPS files with 300dpi resolution at resubmission; please read our figure guidelines for more information on our requirements: http://journals.plos.org/plosmedicine/s/figures. While revising your submission, please upload your figure files to the PACE digital diagnostic tool, https://pacev2.apexcovantage.com/. PACE helps ensure that figures meet PLOS requirements. To use PACE, you must first register as a user. Then, login and navigate to the UPLOAD tab, where you will find detailed instructions on how to use the tool. If you encounter any issues or have any questions when using PACE, please email us at PLOSMedicine@plos.org.

FIGURES AND TABLES

SUPPLEMENTARY MATERIAL

REFERENCES

STUDY TYPE-SPECIFIC REQUESTS

* PLOS Medicine requires that all trials be prospectively registered in one of registries recognized by WHO. Please ensure that study registration details are included in the Methods section.

* Please structure the Methods section using the following sub-headings: Study design and participants, Randomization and masking, Procedures, Outcomes, Statistical analysis.

* Please clarify and explain all discrepancies between the paper and protocol. We have pointed out some instances below. If outcomes were not prespecified in the protocol, please define them in the Methods (Outcomes section) as post hoc and explain why they were added. Post-hoc comparisons should be presented as hypothesis generating rather than conclusive.

- The following outcome measure, "The primary outcome in this study is the PRWE score measured after one and two years.", appears to differ between the submitted manuscript and the protocol [and/or trial registry]. Please clarify.

- In the protocol, you have listed "the number of wrist movements measured with Axivity accelerometer" as a secondary outcomes. However, this does not seem to be reported in the submitted manuscript. Please clarify.

- In the manuscript, you listed grip strength and adverse events as secondary outcomes, which appears to differ from the protocol. Please clarify.

* Please ensure that all prespecified outcomes (primary, secondary, and exploratory) are listed in the Methods/Outcomes section and indicate whether there are outcomes that are not presented in the current report.

* Please specify the dates (Month Day, Year) during which study enrollment and follow up occurred.

* Please include absolute numbers wherever you report percentages; eg, n/N (%)

* Please present the safety data for the study including numbers of specific events and whether or not adverse events are thought to be related to treatment. AEs should be reported in the abstract, per CONSORT and CONSORT-Harms.

* Please complete the CONSORT checklist (https://www.equator-network.org/reporting-guidelines/consort/) and ensure that all components of CONSORT are present in the manuscript, including how randomization was performed, allocation concealment, blinding of intervention, definition of lost to follow-up, power statement. When completing the checklist, please use section and paragraph numbers, rather than page numbers.

* Please report your abstract according to CONSORT for abstracts, following the PLOS Medicine abstract structure (Background, Methods and Findings, Conclusions) https://www.equator-network.org/reporting-guidelines/consort-abstracts/

* If your trial had to undergo important modifications in response to extenuating circumstances, please complete the CONSERVE-CONSORT checklist and provide in your Supporting Information; (https://www.equator-network.org/reporting-guidelines/guidelines-for-reporting-trial-protocols-and-completed-trials-modified-due-to-the-covid-19-pandemic-and-other-extenuating-circumstances-the-conserve-2021-statement/). When completing the checklist, please use section and paragraph numbers, rather than page numbers.

* In keeping with our commitment to Open Science, please include the study protocol document and analysis plan (including any amendments) as Supporting Information to be published with the manuscript if accepted.

* Please note that PLOS Medicine requires prospective, public registration of a data sharing plan (as part of mandatory clinical trials registration) for all clinical trials that began enrollment on or after January 1, 2019, in accordance with ICMJE requirements.

---

## [Decision Letter · Decision Letter 2]

14 Aug 2025

Dear Dr. Hevonkorpi,

Thank you very much for re-submitting your manuscript "DRIFT trial: Nonoperative treatment vs volar locking plating for distal radius fracture in patients aged 65 years or older: A randomized controlled trial" (PMEDICINE-D-25-02233R2) for review by PLOS Medicine.

Thank you for your detailed response to the reviewers' and editors’ comments. I have discussed the paper with my colleagues, and it has also been seen again by two of the original reviewers. The changes made to the paper were satisfactory to the reviewers. As such, we intend to accept the paper for publication, pending your attention to the reviewers' and editors' comments below in a further revision. When submitting your revised paper, please once again include a detailed point-by-point response to the editorial comments.

The remaining issues that need to be addressed are listed at the end of this email. Any accompanying reviewer attachments can be seen via the link below. Please take these into account before resubmitting your manuscript: ********

In revising the manuscript for further consideration here, please ensure you address the specific points made by each reviewer and the editors. In your rebuttal letter you should indicate your response to the reviewers' and editors' comments and the changes you have made in the manuscript. Please submit a clean version of the paper as the main article file. A version with changes marked must also be uploaded as a marked up manuscript file. Please also check the guidelines for revised papers at http://journals.plos.org/plosmedicine/s/revising-your-manuscript for any that apply to your paper.

We ask that you submit your revision within 1 week (Aug 21 2025). However, if this deadline is not feasible, please contact me by email, and we can discuss a suitable alternative.

Please do not hesitate to contact me directly with any questions (atosun@plos.org).

We look forward to receiving the revised manuscript.

Sincerely,

Alexandra Tosun, PhD

Senior Editor 

PLOS Medicine

plosmedicine.org

Comments from Reviewers:

Reviewer #2: The comments have been adequately addressed in the revision.

Reviewer #3: Thank you to the authors for addressing my previous comments well. My only remaining suggestion would be for the p-value for the primary endpoint that were added to the results to be included in the abstract also.

James Wason

********

Requests from Editors:

GENERAL

* Please confirm that your title complies with to PLOS Medicine's style. Your title must be nondeclarative and not a question. It should begin with main concept if possible. "Effect of" should be used only if causality can be inferred, i.e., for an RCT. Please place the study design ("A randomized controlled trial," "A retrospective study," "A modelling study," etc.) in the subtitle (ie, after a colon).

Suggestion: Nonoperative treatment vs volar locking plating for distal radius fracture in patients aged 65 years or older (DRIFT trial): A randomized controlled trial

* Statistical reporting: Please revise throughout the manuscript, including tables and figures.

- Please report statistical information as follows to improve clarity for the reader ""22% (95% CI [13,28]; p</=)"".

- Please separate upper and lower bounds with commas instead of hyphens as the latter can be confused with reporting of negative values.

- Please repeat statistical definitions (HR, CI etc.) for each set of parentheses.

* Please ensure that all abbreviations are defined at first use throughout the text (including statistical abbreviations).

* Please ensure that tables and figures, including those in supplementary files, are appropriately referenced in the main text.

* Please review your text for claims of novelty or primacy (e.g. 'for the first time' or ‘novel’) and remove this language. 

* Please confirm that any use of statistical terms (such as trend or significant) are supported by the data, and if not please remove them. The term trend should be used only when the test for trend has been conducted.

* Please define all acronyms used in each figure or table in its corresponding legend.

* Please review your text for claims of novelty or primacy (e.g. 'for the first time') and remove this language. 

* CONSERVE-Checklist: Please change the page numbers to sections and paragraphs, as you did in the CONSORT 2025 checklist.

* In trials, there is usually a distinction in the language in terms of causal vs associational for primary and secondary trial outcomes. It would be beneficial to use associational language in the discussion and other sections for secondary outcomes. Please check and revise, if necessary. 

ABSTRACT

* Please confirm that your abstract complies with our requirements, including providing all the information relevant to this study type https://journals.plos.org/plosmedicine/s/submission-guidelines#loc-abstract

* We suggest specifying ‘Nordic countries’.

* When reporting age, please add a unit, such as ‘years’. Please revise throughout. 

* Please ensure that all numbers presented in the abstract are present and identical to numbers presented in the main manuscript text.

* In the abstract, please include the important dependent variables that are adjusted for in the analyses.

* When presenting the results, please make sure readers know what the comparator is.

* l.54ff: We recommend removing the colons and restructuring the sentences as complete ones.

* Please include the clinical trial registry number in the abstract.

AUTHOR SUMMARY

* Given that the estimated mean effect for PRWE at 12 months was smaller than the predefined MCID, we suggest revising the statement that the difference may be clinically significant.

INTRODUCTION

* Please ensure that the Introduction ends with a clear description of the study question or hypothesis.

METHODS AND RESULTS 

* l.160ff: Please revise using full sentences rather than colons and bullet points. Please revise throughout.

* “Research permits were obtained from the local ethics committees and hospital districts prior to the commencement of the trial. – please provide approval numbers for each ethics committee.

* “The DRIFT trial was conducted in five trauma centers in the Nordic Countries.” – please specify. 

* “Low energy intra-articular or extra-articular dorsally displaced DRF within 3 cm of the radiocarpal joint” – Why was the information "diagnosed with lateral and posterior-anterior radiographs in the ER" removed from the main text, despite being written in the SAP?

* l.205, “AO-type A or C” – have you defined this anywhere?

* Under ‘Outcomes’, please report that the 24- months results will be reported later as explained in your rebuttal. A reminder that we require you to report all prespecified outcomes (primary, secondary, and exploratory) outlined in the protocol and indicate whether there are outcomes that are not presented in the current report.

* Figure 1: Please note that in the flowchart you have classified that 3 months time point as primary outcome which is not 100% correct. 

* Figure 2: Would you consider aligning the y-axis of both graphs?

* Tables: We suggesting inserting gridlines for better readability. 

DISCUSSION

* “the Nordic setting of the trial” – Given our global audience, this may not be clear to all readers. Please revise.

* “Our trial is also the first RCT to compare the operative..” – please remove claims of primacy or add ‘to our knowledge’ (or similar).

General Editorial Requests

---

## [Editor Report · Decision Letter 3]

21 Aug 2025

Dear Dr Hevonkorpi, 

On behalf of my colleagues and the Guest Academic Editor, Ian Harris, I am pleased to inform you that we have agreed to publish your manuscript "Nonoperative treatment vs volar locking plating for distal radius fracture in patients aged 65 years or older (DRIFT trial): A randomized controlled trial" (PMEDICINE-D-25-02233R3) in PLOS Medicine.

I appreciate your thorough responses to the reviewers' and editors' comments throughout the editorial process. We look forward to publishing your manuscript, and editorially there is only one remaining point that should be addressed prior to publication. We will carefully check whether the change has been made. If you have any questions or concerns regarding these final requests, please feel free to contact me at atosun@plos.org.

Please see below the minor point that we request you respond to:

* Discussion: “In addition, the trial was conducted in the Nordic countries with rather good healthcare resources on a global scale.” – please define ‘Nordic countries’.

Before your manuscript can be formally accepted you will need to complete some formatting changes, which you will receive in a follow up email (including the editorial request above). Please be aware that it may take several days for you to receive this email; during this time no action is required by you. Once you have received these formatting requests, please note that your manuscript will not be scheduled for publication until you have made the required changes.

PRESS

Sincerely, 

Alexandra Tosun, PhD 

Senior Editor 

PLOS Medicine